# Safety Practices and Associated Factors among Healthcare Waste Handlers in Four Public Hospitals, Southwestern Ethiopia

**Sisay Ketema** [1,*] **, Abayneh Melaku** [2] **, Habtamu Demelash** [3] **, Meseret G/Mariam** [1] **, Seblework Mekonen** [2] **, Taffere Addis** [2,*] **and Argaw Ambelu** [2]

1 Department of Public Health, Mizan Aman College of Health Science, Addis Ababa P.O. Box 1138, Ethiopia; gebrumesimes@gmail.com
2 Ethiopian Institute of Water Resources, Addis Ababa University, Addis Ababa P.O. Box 1138, Ethiopia; aambelu@yahoo.com (A.A.)
3 Department of Public Health, College of Health Science, Debre Tabor University, Debre Tabor P.O. Box 272, Ethiopia
* Correspondence: sisayketema8@gmail.com (S.K.); taffere.addis@aau.edu.et (T.A.)

**Abstract:** Occupational safety is a critical concern for disease prevention and control at healthcare facilities. Medical waste handlers, in particular, are those most exposed to occupational hazards among healthcare workers. Therefore, this cross-sectional study was conducted to evaluate safety practices and associated factors among healthcare waste handlers in four public hospitals, southwest Ethiopia from 15 March to 30 May 2022. The study included 203 healthcare waste handlers. The data were collected using an interviewer-administered questionnaire and observational checklists. The overall performance of occupational safety practices among healthcare waste handlers was 47.3% (95%CI; 40.3, 54.2). Waste handlers with an educational status of secondary and above (AOR 4.95; 95%CI 2.13, 11.50), good knowledge of infection prevention and safety practices (AOR 4.95; 95%CI 2.13, 11.50), training in infection prevention and safety practices (AOR 2.57; 95%CI 1.25, 5.29), and adequate access to safety materials (AOR 3.45; 95%CI 1.57, 7.60) had significantly better occupational safety practices than their counterparts. In general, medical waste handlers' occupational safety practices were found to be inadequate. Waste handlers' knowledge of safety measures and training, educational level, and availability of safety materials were predictors of safe occupational practices. Therefore, appropriate strategies and actions are needed to ensure the safe occupational practices of healthcare waste handlers.

**Keywords:** occupational safety; medical waste handler; healthcare waste management; safety practice; infection prevention and control; Ethiopia



## 1. Introduction

Healthcare facilities generate hazardous waste that can endanger waste handlers, patients, health workers, visitors, the general public, and the environment, including soil and water bodies [1]. To prevent occupational health risks for people who handle medical waste, strict adherence to universal precautions, as well as appropriate segregation, collection, storage, transportation, treatment, and disposal of healthcare waste, is needed [2]. The Centre for Disease Prevention and Control recommends that everyone in a healthcare facility adhere to universal precautions consistently and correctly [3]. These precautions include proper hand washing, and the wearing of face masks, protective eye shields, hand gloves, aprons and/or gowns, and safety boots, as well as other safety measures such the proper segregation and disposal of medical waste to prevent occupational diseases and injuries in the workplace [4,5]. Appropriate personal protective equipment must be worn at all times when working with healthcare waste; it should be confirmed to be in good condition before each use, and any equipment needing replacement or maintenance should be reported to management [6]. However, compared with healthcare workers,

waste handlers are those most exposed to occupational hazards such as infectious diseases and sharps injuries in healthcare settings that lack sound waste management systems and provision of personal protective equipment [7]. For example, more than 40% of healthcare waste handlers in a tertiary care hospital of Mangalore (India) had been exposed to needle sticks and other waste injuries [8].

Today, massive investments are being made worldwide towards the expansion of public and private healthcare facilities, resulting in the production of millions of tons of hazardous waste, including infectious waste. Despite the hazardous nature of medical waste, healthcare facilities have placed less focus on safe medical waste management, and problems are prevalent in developing countries, including in Ethiopia [9]. Additionally, healthcare facilities in developing countries lack data on the generation rate, composition, and characterization of medical waste, making the implementation of safe management systems challenging and contributing to an increase in the incidence of waste-management-related health risks among medical waste handlers. Less than 33% of healthcare facilities in the least developed nations have basic healthcare waste management services [10]. Medical waste handlers are more vulnerable when handling, segregating, transporting and disposing of waste at a facility, because the waste is carelessly disposed of everywhere on the healthcare premises, in total disregard of duty of care principles during patient handling, diagnosis, and treatment [4,11,12].

Hepatitis B virus (HBV) infection is the most common health problem among healthcare workers. This disease is more common in low- and middle-income countries, mainly in Africa and Asia, where resources such as personal protective equipment and other prevention measures, including hepatitis B virus (HBV) vaccination for the workers at service delivery, are limited [13]. As vaccination is the main HBV prevention strategy, the CDC recommends a three-dose vaccine that gives protection of more than 90% after the third dose for all healthcare workers [14]. However, estimated full-dose HBV vaccination coverage among healthcare workers in Ethiopia is 20.04% [15]. In Ethiopia, HBV vaccination has been added for children under one year of age in its expanded universal immunization program as of 2007, but not for healthcare workers, yet.

Additionally, healthcare waste handlers were considered as the ancillary group, despite the fact that they play a pivotal role for the success of medical waste management, and prevention of nosocomial infections and accidents [6]. As a result, healthcare services managers give less priority to the prevention of direct exposure of healthcare waste handlers with infected bodily fluid, used hypodermal needles, cleaning agents, X-ray, and dust containing pathogenic microorganism, specimens, laboratory reagents, and cytotoxic drugs. Hence, healthcare waste management workers are a group more at risk than other healthcare staff members, unless the necessary measures are in place to protect them from serious health risks and injuries.

Waste handlers who have been exposed to infected bodily fluids can acquire blood borne viruses such as Human Immunodeficiency Virus (HIV), as well as hepatitis B and C infections [16,17]. A person who has experienced a single needle stick injury from a needle used for an infected patient has risks of 30%, 1.8%, and 0.3% of becoming infected with HBV, the hepatitis C virus (HCV), and HIV, respectively [18,19]. Hence, solving the problems of inadequate personal protective devices and the lack of awareness regarding the proper management of healthcare waste should be the top priority for healthcare infection prevention and control. Hepatitis B virus infection among medical waste handlers was high compared to among non-medical waste handlers [20]. Surprisingly, HBV was detected in 52 (20.4%) medical waste collectors in public healthcare facilities in eastern Ethiopia [17]. Another one-year retrospective study on the same region also found that 75 (30%) medical waste collectors had been exposed to sharp injuries, and 43.8% had been exposed to blood and bodily fluid while handling healthcare waste [16]. A study conducted in hospitals in Addis Ababa revealed that the availability and accessibility of safety materials were determinant factors for safe occupational practices among waste handlers [21]. Along with

the proper handling and disposal of waste, appropriate on-site storage should be available to protect the health of waste handlers.

Although there have been several studies on the occupational safety practice of healthcare waste management workers in hospitals in Ethiopia, there have been no studies performed in the South West Region of Ethiopia. Moreover, studies on the safety of healthcare waste handlers since the decline in COVID-19 incidence are scarce in Ethiopia, even though studying the trend in the aftermath is essential. Hence, this study aims to assess occupational safety practices and associated factors among healthcare waste handlers in four public hospitals in the South West Ethiopia Peoples' Region. This region was founded by splitting it off from the Southern Nations, Nationalities, and Peoples' Region on 23 November 2021 following a referendum. This study provides baseline data for the newly established region in order to be able to plan and implement a contextualized sustainable healthcare waste management system that promotes the occupational safety practices of waste handlers, thereby playing a pivotal role in the optimization of infection and control programs in healthcare facilities.

## 2. Materials and Methods

### 2.1. Study Area

The study was conducted in the South West Ethiopia Peoples' Region, a regional state in southwestern Ethiopia. This region consists of the Keffa, Sheka, Bench Sheko, Dawro, and West Omo Zones, and the Konta Special District. The region has an area of about 39,400 square kilometers and an estimated population of 2.3 million. This study was conducted at four public hospitals, namely Mizan Tepi University Teaching Hospital, Gebretsadik Shawo General Hospital, Tepi General Hospital, and Chena Primary Hospital. These hospitals serve around 3.5 million people.

### 2.2. Study Design and Period

An institutionally based cross-sectional study design was implemented from 15 March to 30 May 2022.

### 2.3. Study Population

The study population included all healthcare waste handlers at Mizan Tepi University Teaching Hospital, Gebretsadik Shawo General hospital, Tepi General Hospital, and Chena Primary Hospital. The numbers of waste handlers at Mizan Tepi University Teaching Hospital, Gebretsadik Shawo General hospital, Tepi General Hospital, and Chena Primary Hospital were 90, 68, 30, and 22, respectively, making a total of 210 healthcare waste handlers.

### 2.4. Sample Size and Sampling Techniques

All healthcare waste management workers in the selected hospitals were included in order to assess occupational safety practice. All healthcare waste management workers were enrolled to participate in the study. A list of names of the waste handlers was obtained from the human resources department of each hospital. The expected number of healthcare waste handlers to be included in this study was 210.

### 2.5. Inclusion and Exclusion Criteria

All healthcare waste management workers who were assigned as waste handlers in the hospitals and who were on the job at the time of data collection were included in the study. For those waste handlers who were seriously ill at the time of data collection and having difficulties responding to the questions, we waited until they had recovered from their illness to take their responses. Additionally, those waste handlers who were not on duty for organizational or personal reasons, including health conditions at the time of the survey, and where it was difficult to obtain their responses before the end of the data collection period, were excluded from the study. In total, only seven healthcare waste

handlers were excluded from the study, leading to a response rate of 96.7%. Recruitment of waste handlers was conducted with the support of administrators and environmental health experts at each hospital.

*2.6. Study Variables*

2.6.1. Dependent Variables

- Safety practice.

2.6.2. Independent Variables

- Socio-demographic factors: Age, sex, marital status, education status, monthly income, service year, and family size.
- Behavioral factors: Smoking tobacco, chewing khat, and alcohol consumption.
- Job characteristics: Duration in the job, number of work hours, nature of employment, training before commencing job, job rotation, handling of waste (manual handling or use of equipment), place of handling waste, type of work (collection of waste, collection and segregation).
- Availability and utilization of personal protective equipment: Use of safety measures (mask, gloves, gumboots, gown, and eye goggles).
- Personal hygiene: Habit of washing oneself and changing clothes after work, taking lunch/snacks during working hours.

*2.7. Data Collection Tools and Procedures*

Data were collected using an interviewer-administered questionnaire and observational checklist adapted from the WHO and similar studies.

2.7.1. Interviewer-Administered Questionnaire

An interviewer-administered questionnaire was used to assess the knowledge and attitude of the study participants regarding their occupational safety. Moreover, safety practices related to personal hygiene, medical waste handling, and the utilization of personal protective equipment, the availability of safety materials, and vaccination status were collected using a questionnaire with further crosschecking by observation. The questionnaire was developed after reviewing different studies [22–27], and modifications were also made to fit with our research objectives. The questionnaire contained both open- and closed-ended questions. The data were collected through face-to-face interviews by trained data collectors with bachelor's degrees in environmental health or public health.

2.7.2. Observational Checklist

An observational checklist was used to identify the utilization and practice of safety measures by waste handlers in the workplace. Additionally, data collectors assessed the overall waste disposal and segregation practices of health facilities using the observational checklists adapted from Mengiste et al. [17] and Tekle et al. [21]. The occupational safety practices that were assessed using the observation checklist included hand washing, personal protective equipment utilization, waste segregation, disinfection of infectious waste matters, availability of proper dust bins for waste storage, and availability of safety guidelines and protocols, each of which determines the risk of exposure to occupational hazards [24].

*2.8. Occupational Safety Knowledge, Attitude, and Practices Operational Definitions*

2.8.1. Knowledge of Infection Prevention and Safety Measures

Good: those who provided correct answers for more than or equal to 50% of the questions that were used to assess the waste handlers' knowledge of safety measures; otherwise, they were considered to have poor knowledge of safety measure [21].

### 2.8.2. Attitude towards Infection Prevention and Safety Measures

Good: those who provided correct answers for more than or equal to 50% of the questions that were used to assess the attitude of waste handlers towards safety measures; otherwise, they were considered to have a poor attitude [21].

### 2.8.3. Safety Practice

Good: those who provided correct answers to and/or were verified by observation to know more than or equal to 50% of the questions that were used to assess the practices of waste handlers; otherwise, they were considered to exhibit poor practice [21].

### 2.9. Data Quality Management

The questionnaire was pre-tested on 5% of the sample of waste management workers at the Mizan Health center and amended based on their feedback. Data collectors and supervisors were trained for two days on the contents of the questionnaire and COVID-19 prevention practice before proceeding to actual data collection. The principal investigator contacted data collectors and supervisors on a daily basis to identify and solve the challenges faced during data collection. Those waste management workers who were not on duty on the day of data collection were given an appointment in the following days to give their responses to questions.

### 2.10. Data Analysis and Interpretation

The collected data were cheeked for their completeness, coded and entered into Epi data version 4.2. Then, the data were exported to SPSS version 23 for data management and analysis. Descriptive results are summarized using means, frequencies, and percentage. Binary logistic regression analysis was employed to select potential predictor variables for safe waste handling and management practice for multiple logistic regression analysis setting $p < 0.2$. Statistical significance for the adjusted odds ratio (AOR) in multivariate logistic regression analysis was determined at $p < 0.05$ and 95% CI.

### 2.11. Ethical Considerations

The study was conducted after obtaining an ethical clearance letter from the Ethical Review Committee of Mizan Aman College of Health Science (Ref. No. MA-HSC/ERC/0040/2022, dated 15 February 2022). Additionally, the Health Science College wrote a letter stating the relevance and objective of the study to selected hospitals to cooperate in the recruitment of participants and smoothing the data collection process. Data from each healthcare waste handler was collected after obtaining verbal informed consent. Privacy and confidentiality were maintained throughout the study period by excluding personal identifiers during data collection.

## 3. Results

### 3.1. Socio-Demographic Characteristics of Study Participants

Among the 210 healthcare waste management workers, 203 (96.7%) participated in the study. Most respondents were females (91.6%), having a mean age of 26.94 years, were married (52.7%), with an educational status of secondary or above (70%), had a monthly income of above ETB Birr (64.5%) and a family size $\leq 5$ (74.4%), was working in a medical service provision department (94.1%) with greater risk, had $\leq 5$ years' experience as a waste handler (76.8%), highlighting the need for intensive training on occupational safety, worked $\leq 8$ h per day (95.1%), and had no medical checkups (72.9%) (Table 1).

**Table 1.** Socio-demographic characteristics of studied participants in four public hospitals in South-western Ethiopia, 2022.

| Variables | Category | Frequency | Percent (%) |
|---|---|---|---|
| Sex | | | |
| | Male | 17 | 8.4 |
| | Female | 186 | 91.6 |
| Age | | | |
| | ≤25 | 100 | 49.3 |
| | 26-30 | 62 | 30.5 |
| | 31–35 | 21 | 10.3 |
| | >35 | 20 | 9.9 |
| | Mean ± SD | 26.9 ± 6.3 | |
| Marital status | | | |
| | Married | 107 | 52.7 |
| | Single | 59 | 29.1 |
| | Widowed | 12 | 5.9 |
| | Divorced | 11 | 5.4 |
| | Separated | 14 | 6.9 |
| Educational status | | | |
| | Primary | 61 | 30.0 |
| | secondary and above | 142 | 70.0 |
| Salary | | | |
| | ≤1500 | 72 | 35.5 |
| | >1500 | 131 | 64.5 |
| Family size | | | |
| | ≤5 | 151 | 74.4 |
| | >5 | 52 | 25.6 |
| Work department | | | |
| | Emergency unit | 27 | 13.3 |
| | Pediatric ward | 31 | 15.3 |
| | Operation room | 18 | 8.9 |
| | Administrative office | 12 | 5.9 |
| | Outpatient department | 31 | 15.3 |
| | Surgical ward | 22 | 10.8 |
| | Maternity ward | 34 | 16.7 |
| | Medical ward | 16 | 7.9 |
| | Laboratory unit | 12 | 5.9 |
| Work experience | | | |
| | ≤5 years | 156 | 76.8 |
| | >5 years | 47 | 23.2 |
| Working hours | | | |
| | ≤8 h | 193 | 95.1 |
| | >8 h | 10 | 4.9 |
| Work shift | | | |
| | No | 45 | 22.2 |
| | Yes | 158 | 77.8 |
| Medical checkup taken | | | |
| | No | 148 | 72.9 |
| | Yes | 55 | 27.1 |

SD: Standard deviation.

*3.2. Safety Practices*

From our findings, 96 (47.3%) of the healthcare waste handlers were found to have good overall safety practices. Among the safety measures taken by respondents to protect their health, washing hands at critical times, washing work clothes daily, and having received the hepatitis B vaccine were 86 (42%), 50 (24.6%), and 77 (37.9), respectively. The number of waste handlers who were trained in infection prevention and safety measures was low (80, 39.4%) (Table 2).

**Table 2.** Safety practices among healthcare waste handlers at four public hospitals in Southwestern Ethiopia, 2022.

| Safety Practices | Responses | Frequency | % |
|---|---|---|---|
| Do you of often wash your hands after work? | Yes | 86 | 42.4 |
| | No | 117 | 57.6 |
| Do you often change your work clothes immediately after work? | Yes | 99 | 48.7 |
| | No | 104 | 51.3 |
| Do you wash your work clothes every day after work? | Yes | 50 | 24.6 |
| | No | 153 | 75.4 |
| Have you ever received training regarding infection prevention? | Yes | 80 | 39.4 |
| | No | 123 | 60.6 |
| Have you taken the hepatitis B virus vaccine? | Yes | 77 | 37.9 |
| | No | 126 | 62.1 |
| Have you taken the tetanus toxoid vaccine? | Yes | 36 | 17.7 |
| | No | 167 | 82.3 |
| Do you use any personal protective equipment while you are on duty? | Yes | 79 | 38.9 |
| | No | 124 | 61.1 |
| Do you disinfect/decontaminate reusable cleaning devices after each use? | Yes | 35 | 17.2 |
| | No | 168 | 82.8 |
| Do you collect infectious medical waste from the service area within 24 h? | Yes | 134 | 66.0 |
| | No | 69 | 34 |
| Do you always separately transport medical waste in a segregated manner? | Yes | 39 | 19.2 |
| | No | 164 | 80.8 |
| Do you always close medical waste containers during transport? | Yes | 68 | 33.4 |
| | No | 135 | 66.6 |
| Do you clean your hands with alcohol after coming into contat with dirty surfaces? | Yes | 35 | 17.2 |
| | No | 168 | 82.8 |
| Overall safety | Yes | 96 | 47.3 |
| | No | 107 | 52.7 |

*3.3. Source of Information for Safety Measures*

Information regarding safety measures is crucial in order for workers to be aware of the dangerous nature of their work, and to take care of their health while performing their routine tasks. Participants responded that their major sources of information were training (35%), friends (22%), television (6%), and radio (4%). Healthcare waste handlers who were unaware of infection prevention and safety measures accounted for 19% of total healthcare waste handlers.

*3.4. Healthcare Waste Handlers' Satisfaction on the Job*

Job satisfaction contributes significantly to a firm, resulting in improved service delivery, while unsatisfied workers can negatively affect organizational outcomes. The medical waste handlers in four hospitals responded that the lack of safety materials (28%), supportive supervision (25%), and low salary (25%) were among the major causes of poor satisfaction.

*3.5. Factors Associated with Safety Practice of Healthcare Waste Handlers*

Bivariate and multivariate logistic regression analysis showed that safety practice was significantly associated with age $\leq 25$ (AOR 7.46 (1.77, 31.43), secondary and above educational status (AOR 4.95 (2.13, 11.50), good knowledge of infection prevention and safety (AOR 4.95 (2.13, 11.50), training on infection prevention and safety measures (AOR 2.57 (1.25, 5.29), and availability of safety materials (AOR 3.45 (1.57, 7.60) (Table 3).

**Table 3.** Factors associated with safety practice among healthcare waste handlers in four public hospitals in Southwestern Ethiopia, 2022.

| Variable | Category | Safety Practice | | COR (95%CI) | AOR (95%CI) |
|---|---|---|---|---|---|
| | | No | Yes | | |
| Age | | | | | |
| | ≤25 | 38 | 62 | 6.52[2.03, 20.97] * | 7.46[1.77, 31.43] * |
| | 26–30 | 41 | 21 | 2.04[0.60, 6.90] | 2.44[0.575, 10.43] |
| | 31–35 | 13 | 8 | 2.46[0.604, 10.04] | 2.75[0.520, 14.50] |
| | >35 | 16 | 4 | 1 | 1 |
| Educational status | Primary | 97 | 61 | 1 | 1 |
| | Secondary and above | 11 | 34 | 4.91[2.31, 10.42] * | 4.95[2.13, 11.50] * |
| Experience | ≤5years | 79 | 77 | 1.57[0.806, 3.05] | 0.930[0.390, 2.21] |
| | >5years | 29 | 18 | 1 | 1 |
| Knowledge of infection prevention and safety measures | Poor | 95 | 71 | 1 | 1 |
| | Good | 13 | 24 | 2.47[1.17, 5.18] * | 4.95[2.13, 11.50] * |
| Attitude on infection prevention and safety measures | Poor | 89 | 80 | 1 | 1 |
| | Good | 19 | 15 | 0.878[0.419, 1.84] | 0.997[0.410, 2.42] |
| Training on infection prevention and safety measures | No | 84 | 57 | 1 | 1 |
| | Yes | 24 | 38 | 2.33[1.26, 4.30] * | 2.57[1.25, 5.29] * |
| Availability of safety materials | No | 89 | 64 | 1 | 1 |
| | Yes | 19 | 31 | 2.269[1.17, 4.36] * | 3.45[1.57, 7.60] * |

COR = crude odds ratio; AOR = adjusted odds ratio. * Statistically significant at $p < 0.05$.

### 3.6. Management of Injuries

Injuries are highly prevalent in the work area, particularly in medical settings, due to the nature of the work. Thus, risk mitigation measures should be in place to minimize accidental exposure to hazards. Participants reported that they washed their hands with soap and water (20%), rubbed their hands with alcohol or iodine (13%), and were tested for HIV (8%), while overall injury management actions were 33%

### 3.7. Availability of Safety Materials for Healthcare Waste Handlers

Hospitals are high-risk areas, with both biological and non-biological hazards being common, resulting in injury to individuals unless universal and transmission-based precautions are adhered to. Basically, the availability of safety materials should not be a limiting factor for the protection of workers' health. But this seems to be the case, with only 73% of medical waste handlers having work clothing, and 33%, 33%, and 2% having access to heavy-duty gloves, boots, and masks, respectively.

## 4. Discussion

In this study, we attempted to determine the status of occupational safety practices and associated factors among healthcare waste handlers in public hospitals situated in the South West Region of Ethiopia. In Ethiopia, despite highly infectious waste being generated in healthcare facilities, unsafe waste handling and management practices in public health facilities is one of the most typical problems experienced in the country [21,24,28]. In this study, it was revealed that the overall prevalence of good waste handling practices among healthcare waste handlers was 47.3%, which is low. These results are comparable with those of studies conducted among healthcare waste management workers in governmental hospitals in Addis Ababa (44.1%) [21], in eastern Ethiopia public health facilities in (42.3%) [29], and in Kathmandu Metropolitan City in Nepal (45.8%) [30]. However, our findings were

higher than those in studies conducted among healthcare professionals in small and remote towns in the South Omo Zone of southwest Ethiopia (29.3%) [28], and among healthcare waste management workers in Bengkulu city in Indonesia (35.21%) [31], the Bale Zone in southeast Ethiopia (36.8%) [32], and Ondo City in southwest Nigeria (6.1%) [33]. A possible explanation for the differences between the results of this study and those of the study conducted in the South Omo Zone and the Bale Zone of Ethiopia might be related to the distance from the national capital and the level of urbanization. That is, healthcare waste handlers residing close to the capital and to more urbanized settings (i.e., this study) had better safety practice in healthcare waste handling and management performance when compared with those from smaller and more remote towns, which is probably related to the availability of information and safety materials. This study highlighted that safety and infection prevention and control practices in healthcare facilities are lagging behind the 2030 Sustainable Development Goals (SDG), which demand strategic action from the government and from health institutions.

Previous studies have reported that healthcare waste handlers having good knowledge of safe waste handling and healthcare waste management had better safety practices in the work place [31,34]. In this study, healthcare waste handlers with good knowledge of safe waste handling practices were 4.95 times more likely to perform their jobs safely than those with poor knowledge, which indicates the importance of safety training among healthcare waste handlers to ensuring health and safety in healthcare facilities. Our results were consistent with those of studies conducted among healthcare waste handlers at governmental hospitals in Addis Ababa, where it was reported that those with good knowledge were 4.7 times more likely to adhere to safe waste handling practices [21], and among healthcare professional in Addis Ababa [35], and hospital workers in the Bale Zone of the Oromia region (Ethiopia), when compared with those who had poor knowledge [32]. One of these studies also ascertained that having good knowledge of infection prevention and safety measures resulted in improved safety measures through the anticipation of risks during work [35]. Conversely, our findings are not consistent with those of a study performed in South India [36]. This difference could be due to the socio-demographic characteristics of the study participants.

Waste handlers with adequate supplies were 3.45 times more likely to have safe practices than those with insufficient supplies. This finding is in line with studies conducted among public health waste handlers in Debre Markos [23] and Addis Ababa [21], and among healthcare workers in the Bale Zone [32]. The provision of personal protective equipment such as gloves, boots, and masks, along with adequate cleaning agents and washing facilities with a continuous water supply, are vital to maintaining the safety and health of healthcare waste handlers [10,20]. Therefore, healthcare administrators, medical directors, and central supply units should place provision of safety equipment and other supplies among their top priorities.

The level of education contributes greatly to the ability of workers to acclimatize to their work environment and act in a safe manner, as education provides basic knowledge regarding the work environment. In this study, it was found that participants with secondary education and higher were 4.95 times more likely to use safety measures to protect their health than their counterparts. Similarly, the study conducted in Alkut City (eastern Iraq) among healthcare workers indicated that staff with higher level of education had a better capacity to practice safe waste handling and implement management measures for the control and prevention of injuries and exposure to hazards such hazardous chemicals, physical agents, and infectious organisms in the work place [34]. Healthcare waste handlers play a critical role in nosocomial infection prevention and injury prevention, because they are involved in the collection, storage and safe disposal of waste that is infectious and hazardous [6]. Therefore, waste handlers who are employed at healthcare facilities should receive basic training on health, safety, and solid waste management, which is not currently the case in Ethiopia, or most probably in other developing countries.

It was revealed in this study that the likelihood of participants using safety measures was significantly associated with their age ($p < 0.05$), with participants with an age less than or equal to 25 years being 7.46 times more likely to exercise safety measures than older age groups. This demonstrates that with increasing age, waste handlers might become more reluctant to practice safety while adapting to work culture and the surrounding conditions. These findings are inconsistent with those of studies performed among public hospital waste management workers in Addis Ababa [21] and among healthcare workers in primary healthcare centers in Alkut City (Iraq) [34]. This may be due to the fact that healthcare waste handlers in remote settings, as is the case in our study area, are probably a mix of youngsters who have completed secondary and college/university education, as compared to more urbanized settings, where the workforce is dominated by those with primary education, as more educated people have more job access, allowing them to shift jobs even after having been employed. Hence, healthcare facilities can promote safe practice among waste handlers through induction training for new workers and in-service training for their existing staff, in addition to hiring people who have undergone basic training on safety and waste management, preferably following completion of secondary education.

In fact, training regarding infection prevention and safety measures for waste handlers has a significant positive impact in terms of protecting them against occupation-related health risks [37]. However, only 80 (39.4%) of the healthcare waste handlers had received training on infection prevention and control. In this study, it was shown that trained healthcare waste handlers were 2.57 times more likely to practice safety measures than untrained individuals. The findings of Dhahir et al. [34] further support the fact that training equips workers with information regarding the nature of the work and safety concerns. Conversely, the findings of this study were not consistent with those of a study conducted among healthcare waste handlers in selected governmental hospitals in Addis Ababa [21]. The impact of training on the likelihood of safe practice in our study was smaller than that reported in the study conducted in Addis Ababa. This might be due to the difference in the sociodemographic characteristics of the participants. Waste segregation (19.2%), handwashing after work (42.4%), disinfecting/decontaminating reusable cleaning devices after each use (17.2%), and vaccination against HBV (37.9%) and tetanus toxoid (17.7%) were rarely practiced safety measures. However, these safety measures are highly relevant for infection prevention and ensuring safety for the waste handlers, health workforce and the community. To improve the safety practices of waste handlers, healthcare managers and environmental health practitioners should scale up induction and the provision of on-the-job training on safety and sustainable waste management practices [38].

## 5. Limitations of the Study

This study did not include healthcare waste handlers in private hospitals and health centers. Additionally, the study lacks the qualitative aspects of safety issues, which are vital for gaining new information and insights.

## 6. Conclusions

It was found in this study that the overall occupational safety practices among healthcare waste handlers in public hospitals in South West Region of Ethiopia were 47.3%, which is low. Waste segregation, handwashing after work, disinfecting/decontaminating reusable cleaning devices after each use, and workers' vaccination against HBV and tetanus toxoid were rarely practiced despite the fact that these are the most important safety measures. Educational status, knowledge of infection prevention and occupational safety measures, training, and the availability of safety materials were the factors that significantly affected healthcare waste handlers' occupational safety practices. Therefore, there should be appropriate implementation strategies to improve the overall occupational safety practices and measures, including pre-service and in-service training, supplying adequate safety materials, and providing the opportunity to scale up the knowledge, attitude, and safe practice among healthcare waste handlers.

**Author Contributions:** S.K. and M.G.: Conceptualization, Methodology, Writing—original draft, review, and editing. A.M. and T.A.: Methodology and data analysis. A.A. and T.A.: Conceptualization, review, and editing. S.M. and H.D.: Writing—original draft, Review, and editing. All authors have read and agreed to the published version of the manuscript.

**Funding:** Mizan Aman College of Health Science funded the data collection of this research. The funding number was Ref. No. MA-HSC/Dean/318/2022, dated 25 February 2022.

**Institutional Review Board Statement:** The study was conducted after receiving an ethical clearance letter from the Ethical Review Committee of Mizan Aman College of Health Science and permission from each studied hospital. Additionally, verbal consent was obtained from each study participant prior to conducting the interview. The confidentiality and privacy of the participants were preserved by removing personal identifiers.

**Informed Consent Statement:** Not applicable.

**Data Availability Statement:** All the relevant data of the study are presented in the manuscript. Raw data can be obtained from the principal investigator upon reasonable request.

**Acknowledgments:** We are grateful to Mizan Aman Health Science College for providing financial and logistic support for this study. Our special thanks go to all hospitals' management bodies for their cooperation, and the participants for their willingness to participate in the study.

**Conflicts of Interest:** The authors declare no conflict of interest.

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
