# Peer review of "Safety Practices and Associated Factors among Healthcare Waste Handlers in Four Public Hospitals, Southwestern Ethiopia"

_safety, 2023_

Round 1

Reviewer 1 Report

first, congratulation for working in this important topic! very relevant paper!!

title: ok

abstract: ok

introduction

here and in the discussion section, you should also cite papers about similar studies around the world (try to include from differente continents, not only Africa);

pag 1, lines 38-42, good point about precautions, however, is very important to include detail about risk and PPE use (not all healthcare activity will need PPE). I considerer this topic very relevant do be presented more cleary!

pag 2 - line 44 ("million tons of infectious waste" you should use hazardous waste (as healthcare facilities also generate chemical, radioactive, etc.)

pag 2, line 69: "Recent pooled prevalence reviewed report showed that 12 month and lifetime needle stick injuries among health care workers was 28.8% and 43.6% respectively" you should include another references about this context! In Brazil, studies showed higher risk data!

pag 2, lines 82-84: "Although, there are few existing studies was addressing occupational safety practice on healthcare waste management workers in hospitals of Ethiopia, there is no study in Southwest region of Ethiopia." you should explore more about your study relevance (not only because there are no studies, but also you are studying more closelly the relationship about risk and healthcare waste management)! you should improve!

Methods 

pag 3, line 102 (2.3.1): "all healthcare worker were enrolled", but you should describe better about ethical condition: how many workers decide do not participate? and how researchers conduted this recruitment stage? describe in the methods section

pag 3, line 100: "seriouly ill" - could you describe more about these cases? is this ill associated to work condition??

as you describe, you used interviews. please report the registration number about approval human research ethics committee (in the methods section).

Results section:

pag 5: Table 1 (not Table 2 as presented!!): as you describe, workers are from diferent work department. How to analyse occupacional risks, considering each depatment generating different types of wastes? yous should explain this in the results section.

pag 5: Table 2 (not Table 3 as presented!!): "Do you use any PPE while you are on duty?" is very important to describe in more details how you observe this information: asking workers or looking the worker during duty?

- also, I recommend to describe this information for all variables that apllied!

pag 6, line 200: where is the Figure 1???

pag 7, line 206: where is the Figure 2???

pag 7: Table 3 (not Table 4 as presented!!): Educational status presented only 45 (11 no and 34 yes) respondents!! why this small number?? this small number did not represent all workers, in my opinion! you should describe better this context and discuss this limited result!!

pag 7, line 220: where is the Figure 3???

pag 7, line 227: where is the Figure 4???

Discussion section: 

pag 8, lines 229-238; first paragraph: important context about African countries, but what is the context around the world?? what percentage we can considerer ideal (even when comparing different context - social, etc., between continent). here, I also expected comparing with results around the world.

pag 8, lines 262-269: only this about age?? there are many variables to discuss! you should discuss more about this. More age not necessary represent work experience... you should try a deep discussion about all variables, and try more references!!

limitation: there are many limitations you shoul include here!! qualitative context, observational interference, sample size not defined, statistical analysis, etc.

Author Response

Dear reviewer 

Thank you so much for providing highly releva t comments on our manuscript. The response are attached for your evaluation. Thank you so much.

Reviewer 2 Report

This article “Safety practices and associated factors among healthcare waste handlers in four public hospitals Southwestern Ethiopia” is a meaningful research topic. This research should be accepted after major revised.

1.     The discussion in this article is too simple. The main content was descriptive statistical analysis. And this manuscript does not conduct test analysis on the items of each questionnaire topic. In the methods, the multiple regression analysis has been addressed, but these results has not been discussed. These should be greatly revised in this manuscript.

2.     Background literature also needs to be added in this manuscript.

3.     “Binary logistic regression analysis was employed to select potential predictor variables to occupational safety practice for multiple logistic regressions analysis setting p<0.2.” Generally, p value should be setting on 0.1 or 0.05.

Author Response

Thank you so much for your relevant comments

 The responses for all reviewers are compiled in one file. Kindly find the responses related to your comments in the file attached.

Reviewer 3 Report

I would like to thank you authors for providing an insightful paper around the safety practices for handling hazardous waste.

I have some comments that I believe addressing them might improve the quality of the paper for publication:

1) I think the text might need to be proofread. For example, in line 38, the wearing of face mask is not consistent with other ones.

2) Please use short and concise sentences: Line 38 to 42 is one sentence or one paragraph?

3) Line 57 to 62 provides no references. Are these the ideas of the writer or are they originated in some literature?

4) Please use quantitative evidence if you have some: Using terms such as mostly considered (line 57), plays a significant role (line 58), this complicates the prevention (how , where, to what scale?)(line 59), were high (how high? double, triple, over the certain concentration?)(line 73). And this is countless over the entire paper. Please read and replace the qualitative terms with some numbers if it is possible.

5-Please add one or two sentences at the end of literature review and introduction (line 86), indicating the probable contribution of your findings. For example, the outcomes of this paper might be helpful for policy maker in local scales to better address the needs of .....

6-section 2.10. Line 170: The collected data was checked for its completeness: What do you mean?

7-In line 180, you mentioned that the majority (almost all) responders are females. Are there any reason behind this? you might wanna justify this. Because it potentially changes the results. For example, the majority of responders are females due to their proportionately high presence in ..... sector (supported by a reference).

8-Line 22: you might want to eliminate "and" before "safety measures".

9-Overall, the materials and method section is not very well designed and miss some of the data explanation. For example, the author repetitively used "AOR" but not even mentioned in method that what does that mean.

10-The authors added some explanation in the results that is not coming from the results and should be eliminated. Only the outcomes of the paper should be included and justified in the "Results and Discussion" section. For example, line 216, Injuries are highly profound (what do you mean by highly profound?) in work area particularly in the medical setting? (is this coming from results or is a general statement?).

Another example: Line 222: Hospitals are the most dangerous areas where both biological and non-biological hazards are common? (what do you mean by the "most dangerous?"), where is it come from? should it be included in the literature review or results. If you included it in the results, then where can I see this statement in your figures and tables?

11-line 239: Please consider removing redundant words and statements. For example line 239: It is obvious (what do you mean by obvious?), that good knowledge (what do you mean by good?) contribute on the effectiveness of safety practice (how contribute to the effectiveness of safety practices)? Please only explain the contribution:

For example, studies showed that increased awareness about the proper solid waste handling might reduces the associated adverse health effects around 50 to 80 percent (references).

12-Please read the text carefully and eliminate the qualitative words as much as possible. There are all over the body of the paper. examples: "great contribution", significantly associated", "highly profound", "critical time", "mostly considered", "prevalent in developing countries", "massive investment"........

Author Response

Dear reviewer 

Thank you so much for your relevant and helpful comments. The responses are attached for all reviwers in one file. Kindly find your part with the same file.

Reviewer 4 Report

Article: „Safety Practices and Associated Factors among Healthcare Waste Handlers in Four Public Hospitals, Southwestern Ethiopia”

Below are my comments on the manuscript. I ask the Authors to refer to the comments and improve their article.

Abstract:

First sentence: „Occupational safety is a fundamental concern for disease prevention and control to workers in healthcare facility particularly medical waste handlers”. In hospitals, the safety of employees also affects the safety of patients, so I think that words – „to workers” should be crossed out. Also, why "particularly"? Due to the nature of their work, medical staff, especially nurses, are more exposed to occupational infections, so instead of in particular it is better to use "including".

„All healthcare waste handlers were included in the study” - i.e. how many?

You don't need two almost the same sentences, just combine them:

„The overall occupational safety practices of healthcare waste handlers were 46.8% (95%CI; 40.3, 54.2)”. and „Generally, the occupational safety practices of medical waste handlers were found to be low (46.8%)”.

Manuscript

Introduction

I think the introduction is written quite correctly. However, I am wondering about the availability of HBV vaccination. In many countries, this vaccination is mandatory for the entire population, or at least for employees of healthcare facilities. Please tell the readers what it is like in developing countries, including Ethiopia.

Results

Independent variables are listed in chapter 2.5.2. However, I do not find an analysis of these variables in the results. Also, could you please explain what some of the variables might mean? For example, is there any research that married people are better at following procedures than single or divorced people? Or smoking cigarettes? These are just examples. If there is no justification for a given variable, then there is no need to include it. Especially since the authors also did not present their impact in the results of the work.

Table 3 - the abbreviations COR and AOR should be explained below the table

Discusion:

Lines 275-276 „Generally, healthcare waste handlers in relatively bigger urban settings had 275 better safe healthcare waste handling (…)” - where does this request come from? the authors did not analyze the variable "city/village" or the variable "more or less urbanized place"

Conclusion

Lines 341-342 „This study found that the overall occupational safety measures among healthcare waste handlers in southwest region of Ethiopia were 46.8%” - where did that number come from? I can't find it in the results.

In general, an average written article, especially the Discussion chapter - I would expect more thought and discussion of the results and conclusions, along with the general conclusion and recommendations for the future. This is a descriptive article, nothing new, but written quite correctly.

Thank you.

Author Response

Response to reviwers

Round 2

Reviewer 2 Report

None

Author Response

Response

Reviewer 3 Report

I am happy that the majority of my comments are somehow addressed. I believe that the paper can be published by its current form

Author Response

Response to reviwer

Reviewer 4 Report

I thank the authors for the growth of the manuscript.